On the interpretability of fuzzy knowledge base systems

http://orcid.org/0000-0003-4439-7583 Camastra Francesco 1
http://orcid.org/0000-0001-5592-7995 Ciaramella Angelo 1 angelo.ciaramella@uniparthenope.it
Salvi Giuseppe 2
Sposato Salvatore 1
http://orcid.org/0000-0002-4708-5860 Staiano Antonino 1
1 Dipartimento di Scienze e Tecnologie, Università degli Studi di Napoli Parthenope , Naples , Italy
2 Scienze Economiche, Giuridiche, Informatiche e Motorie, Università degli Studi di Napoli Parthenope , Nola , Italy
Cirillo Stefano
Electronic publication date: 2024 Dec 3
Publication date: 2024
Volume: 10
Electronic Location ID: e2558
Received 2024 May 18; Accepted 2024 Nov 7
Copyright: © 2024 Camastra et al.
Copyright year: 2024
Copyright holder: Camastra et al.
License: This is an open access article distributed under the terms of the Creative Commons Attribution License, which permits unrestricted use, distribution, reproduction and adaptation in any medium and for any purpose provided that it is properly attributed. For attribution, the original author(s), title, publication source (PeerJ Computer Science) and either DOI or URL of the article must be cited.
License URL: https://creativecommons.org/licenses/by/4.0/

Keywords: Greedy algorithm, Interpretabile and explainable artificial intelligence, NP-hard problem, Fuzzy knowledge base, Rough set theory

Funding: Fondo per la Crescita Sostenibile—Accordi per l’innovazione di cui al D.M. 31 dicembre 2021 e D.D. 18 marzo 2022-CUP B69J23000500005 This work was supported by “Fondo per la Crescita Sostenibile—Accordi per l’innovazione di cui al D.M. 31 dicembre 2021 e D.D. 18 marzo 2022-CUP B69J23000500005” (P.I. Angelo Ciaramella). The funders had no role in study design, data collection and analysis, decision to publish, or preparation of the manuscript.

==============================
In recent years, fuzzy rule-based systems have been attracting great interest in interpretable and eXplainable Artificial Intelligence as ante-hoc methods. These systems represent knowledge that humans can easily understand, but since they are not interpretable per se, they must remain simple and understandable, and the rule base must have a compactness property. This article presents an algorithm for minimizing the fuzzy rule base, leveraging rough set theory and a greedy strategy. Reducing fuzzy rules simplifies the rule base, facilitating the construction of interpretable inference systems such as decision support and recommendation systems. Validation and comparison of the proposed methodology using both real and benchmark data yield encouraging results.

Introduction

In recent years, computational intelligence methods have experienced a great interest in interpretable and eXplainable Artificial Intelligence (XAI) (Mencar & Alonso, 2018; Fernandez et al., 2019). Techniques for XAI can be model-independent (i.e., they can be applied to any AI algorithm) or model-specific (i.e., they can only be applied to a specific AI algorithm). Furthermore, they can be ante-hoc (transparent or “white box/glass box” approaches) or post-hoc (divided into global explanations or local explanations) explanatory methods (Knapič et al., 2021). In particular, ante-hoc methods can be explained by design or inherently explained (referred to as transparent approaches). Transparent approaches are model-specific and include linear and logistic regression, decision trees, k-nearest neighbors, fuzzy inference systems, rule-based learners, general additive models, and Bayesian models. In recent years, fuzzy rule-based systems (FRS) have gained significant attention in explainable AI (XAI) as an ante-hoc approach. The main components of an FRS are a knowledge base (KB) and an inference module. KB contains all fuzzy rules within a rule base (RB) and the definition of fuzzy sets in the database. The inference module includes a fuzzification interface, an inference system, and a defuzzification interface (Di Nardo & Ciaramella, 2021; Camastra et al., 2015). However, it is crucial to emphasize that FRSs need to remain simple and comprehensible, as they are not inherently interpretable (Mendel & Bonissone, 2019; Fernandez et al., 2019). To ensure that fuzzy rule-based systems represent knowledge that is easily understood by humans, several key aspects must be taken into account. These include, among other things, the compactness of the rule base and the semantic clarity of the fuzzy partition (Jara, González & Pérez, 2020; Mendel & Bonissone, 2019). In addition, FRSs must be properly designed to achieve the desired trade-off between accuracy and explainability for the problem at hand. This work presents a fuzzy rule base minimization algorithm based on rough set theory and a greedy strategy. The problem of finding useless attributes for correct classification is NP-hard and it is intractable when the number of attributes is not small. Therefore, algorithms that provide suboptimal approximate solutions need to be explored.

The proposed fuzzy rule base minimization algorithm called Reducing Rules and Conditions (RRC), provides a greedy approximate solution to the problem of identifying and removing attributes that are irrelevant to correct classification from a set of fuzzy rules. RRC roughly consists of five main stages, as described in Fig. 1. To evaluate a rule pattern (a rule pattern of a rule R has an antecedent composed of a subset of the conditions of R and the same consequent of R) in the search phase, a property, i.e., efficiency, is explicitly defined and associated with each rule pattern. The definition of efficiency reflects the main goal of the RRC algorithm, i.e., to search in each subsequent set for the rule patterns with the smallest length of the antecedent, which covers the largest number of rules while having the least overlap with other subsequent sets. The outline of the article is as follows. The section Basics of Rough Set Theory recalls the fundamentals of Rough Set Theory used in the RRC algorithm; in section Reduce Rules and Conditions Algorithm, the RRC algorithm is described; some experimental results are reported in the Experimental Results section, and finally, some conclusions are drawn in the Conclusions section.

Figure 1 Main steps of the RRC algorithm.

Basics of rough set theory

Since the proposed algorithm utilizes approaches and concepts derived from Rough Set Theory (Pawlak, 1982), the foundations of Rough Set Theory are briefly reviewed to improve the readability of the article.

Information system

Following the original approach of Pawlak (1991), the theory of rough sets allows dealing with a certain type of information, i.e., knowledge, associated with a set of objects. In rough sets theory, the knowledge is a collection of facts (or attributes). The attributes are represented by a data table. The rows of the data table correspond to the objects, while the columns of the table represent the attributes associated with the objects. Entries in a row of the data table represent knowledge, i.e., the collection of attributes, associated with the object corresponding to that row. The data table described above is called an information system.

An information system, I, is formally defined as a four-tuple I=(U,Q,V,F), where: U, the universe, is a nonempty finite set of objects;

Q is a finite set of attributes;

V=⋃pVp, where Vp is the domain of the attribute p;

F is an information function giving an attribute value, v∈V, for each pair (object, attribute), and defined as F:U×Q→V, ∀x∈U,∀q∈Q.

Indiscernibility relation

For any set P of attributes, the indiscernibility relation I(P) is defined as follows:

I(P)={(x,y)∈U×U:F(x,a)=F(y,a),∀a∈P}.

If (x,y)∈I(P), then x and y are indiscernibile with respect to P, otherwise they are discernible. The indiscernibility I(P) is an equivalence relation over U. Therefore, it divides U into equivalence classes, i.e., sets of objects that are indiscernible with respect to P. Such a partitioning of U, called classification, is denoted by U/I(P).

Dependency of attributes

For any subset of objects X⊆U and attributes R⊆Q, the R-lower approximation of X, denoted by R_, is given by

R_=⋃{Z∈U/I(R):Z⊆X}.

Given two sets of attributes P and R, a measure of their mutual dependency is the so-called degree of dependency of P on R, γR(P), defined as follows:

γR(P)=|POSITIVER(P)||U|,

where POSITIVER(P)=⋃X∈U/I(P)R_X.

The set POSITIVER(P) is called the positive region of P. This set contains the objects of U that can be classified as belonging to one of the equivalence classes of I(P), using attributes belonging to the set R. The parameter γR(P), taking values in [0,1], represents the fraction of the objects that can be correctly classified.

Significance of attributes

The parameter γR(P) allows defining the significance of an attribute (Modrzejewski, 1993). The significance of an attribute x∈R⊆Q, measures the importance of the attribute x in R in the classification U/I(P). The significance of an attribute, denoted by σxR, is defined as follows:

(1) σxR(P)=γR(P)−γR−{x}(P).

It is worth noting that the meaning thus defined is relative, since it depends on both P and R sets. Thus, an attribute may have a different meaning for different classifications and contexts.

Reduction of attributes

Let I=(U,Q,V,F) be an information system, and let P be a subset of Q (i.e., P⊆Q). The set P is said independent in I, if ∀T⊂P⟹I(P)⊂I(T). The set R⊆P⊆Q is a reduct of P, if it is independent and I(R)=I(P). In other words, any reduct R of a set P classifies the same as P, and the attributes P−R are useless for the correct classification of objects.

Therefore, it can formulate the so-called Reduction of Attributes Problem, which consists in finding the reduct R of a given set of attributes P. The reduction of attributes problem recalls the well-known feature selection problem in machine learning (Hastie, Tibshirani & Friedman, 2009; Ferone & Maratea, 2021; Ferone, 2018; Nardone, Ciaramella & Staiano, 2019), which consists in selecting the features that are relevant (or equivalently removing the irrelevant ones) for classifying data correctly. Feature selection is an NP-hard problem (Davies & Russell, 1994), since it can be polynomially reduced to a VERTEX-COVER problem (Cormen, Leiserson & Rivest, 2009), that is known to be NP-hard. Since each attribute can be polynomially mapped to a feature, the reduction of attributes problem can be polynomially reduced to the feature selection problem, and thus is also NP-hard. This implies that the reduction of attributes problem becomes intractable when the number of attributes is not small.

Reduce rules and conditions algorithm

As shown in the section Reduction of attributes, the problem of finding the useless attributes for the correct classification of objects is intractable when the number of attributes is not small. Therefore, in most real-world applications it is necessary to look for algorithms that provide suboptimal, approximate solutions to the aforementioned problem. The Reduce Rules and Conditions (RRC) algorithm provides a greedy approximate solution to the problem of identifying and removing the attributes, from the set of fuzzy rules, that are irrelevant to the correct classification. Before describing the proposed RRC algorithm, the following definition needs to be introduced.

Definition 1 Let G be a rule, denoted by if A⟹D, where the consequent D is a fuzzy relation and the antecedent A is given by A=⋀iAi, and {Ai}i=1n are fuzzy relations.

A rule pattern P of G is defined as: if A∗⟹D, with A∗=⋀iS(αi,Ai), where αi is a boolean variable and S(αi,Ai) is:

S(αi,Ai)={Aiifαi=1∅otherwise}.

The number of not-null αi, denoted by λ(P), is called the antecedent length of the rule pattern P.

More simply, as shown in Fig. 2, a rule pattern of a rule G, is a rule that has an antecedent composed of a subset of the conditions of G and the same consequent of G.

Figure 2 Example of a rule and its rule patterns.

To evaluate a rule pattern within the algorithm search, the property of the rule pattern, the efficiency, is suitably defined in this work. Specifically, given a rule pattern P and a consequent D, the efficiency of the rule pattern η(P,D), is given by:

(2) η(P,D)=|supΣ⁡(P)∩R(P,D)|(|supΣ⁡(P)|−|supΣ⁡(P)∩R(P,D)|)L+λ(P)K

where supΣ(P) is the superior of set of rules Σ that match the antecedent of the rule pattern P, R(P,D) is the set of rules with consequent D that match the antecedent of the rule pattern P, and λ(P) is the length of the antecedent of the rule pattern P. The optimal values for L and K can be determined by model selection techniques (Hastie, Tibshirani & Friedman, 2009). However, it has been found empirically that typical good values for L and K are 10 and 2, respectively.

The definition of efficiency reflects the RRC algorithm rationale, i.e., for each consequent, searching for rule patterns with the smallest antecedent length, covering the largest number of rules, and having, at the same time, the minimum overlap with other consequents. The greater the number of rules covered by the rule pattern for a given consequent, the better the rule pattern is because it can represent a larger number of rules. Therefore, this value must be proportional to the efficiency, thus justifying the numerator of the Eq. (2).

The denominator expresses the number of rules covered by the rule pattern for other consequents, and the length of the rule pattern, raised to L and K, respectively. These two values must be as small as possible since the overlap with other consequents must be minimal; in addition, the antecedent length of the rule pattern should be as small as possible. In Fig. 3 an example of the efficiency of a rule pattern computed in a knowledge base is presented.

Figure 3 Example of the efficiency of a rule pattern computed in a knowledge base.

Green denotes the attributes of the rule pattern; red denotes the rules that match the antecedent of the rule pattern and have the same consequent; blue denotes the rules that match the antecedent of the rule pattern and have a different consequent.

In practice, RRC takes as input three parameters, R, C, T, namely, the minimum efficiency, the minimum rule coverage, and the number of iterations, and a fuzzy knowledge base F. The algorithm is composed of five steps, which in turn is going to deepen in the next paragraphs: Building decision tables from fuzzy rules;

Sorting attributes by their significance;

Building prefix tree;

Rule pattern searching;

Sorting rule patterns and building the minimum set of rules.

Building decision tables from fuzzy rules

In this stage, each rule of a fuzzy knowledge base, F is represented in a decision system model, where antecedents and consequents of the fuzzy rule are labeled as condition and decision attributes, respectively. To explain the procedure, as an example, consider the following rules:

IF organ_invasiveness IS high AND organ_migration_road IS high ⟹ natural_habitat_colonization2 IS high.

IF organ_invasiveness IS high AND organ_migration_road IS low ⟹ natural_habitat_colonization2 IS medium.

IF organ_invasiveness IS high AND organ_migration_road IS medium ⟹ natural_habitat_colonization2 IS high.

IF organ_invasiveness IS low AND organ_migration_road IS high ⟹ natural_habitat_colonization2 IS medium.

IF organ_invasiveness IS low AND organ_migration_road IS low ⟹ natural_habitat_colonization2 IS low.

IF organ_invasiveness IS low AND organ_migration_road IS medium ⟹ natural_habitat_colonization2 IS medium.

IF organ_invasiveness IS medium AND organ_migration_road IS high ⟹ natural_habitat_colonization2 IS high.

IF organ_invasiveness IS medium AND organ_migration_road IS low ⟹ natural_habitat_colonization2 IS medium.

IF organ_invasiveness IS medium AND organ_migration_road IS low ⟹ natural_habitat_colonization2 IS medium.

Firstly, a label is assigned to each pair (Linguistic Variable, Value), as shown in Table 1.

Table 1 Representation of (Linguistic variable, Value) pairs.

Linguistic variable	Value	Label	
organ_invasiveness	High	H	
organ_invasiveness	Low	L	
organ_invasiveness	Medium	M	
organ_migration_road	High	H	
organ_migration_road	Low	L	
organ_migration_road	Medium	M	
natural_habitat_colonization2	High	H	
natural_habitat_colonization2	Low	L	
natural_habitat_colonization2	High	M	

Then, each pair (Linguistic Variable, Value) in the fuzzy rules is replaced by the corresponding label of Table 1. The resulting decision system is described in Table 2.

Table 2 Decision system resulting from the first stage.

organ_invasiveness	organ_migration_road	natural_habitat_colonization2	
H	H	H	
H	L	M	
H	M	M	
L	H	M	
L	L	L	
L	M	M	
M	H	H	
M	L	M	
M	M	M	

Sorting attributes by their significance

From the decision table, the relevance of each attribute (i.e., for each fuzzy relation) is computed. For this purpose, the relevance of the attribute is measured by its significance, as defined in the Significance of attributes section. Then, the attributes are sorted by their significance (see Fig. 4). The sorting of the attributes allows the implementation of a greedy strategy in the next steps of the algorithm.

Figure 4 The decision table before and after sorting the attributes.

Building prefix tree

In this phase, a prefix tree is built. The prefix tree is built by processing one rule at a time. For each attribute, a node is created, which becomes a child of the node of the previous attribute and a parent of the node of the next attribute. The first attribute becomes the child node of the root tree, i.e., an empty node, since it has no attribute. The tree has the following properties, namely, that all descendants of a node share the prefix associated with the node, and this implies that two rules having the same initial condition, share the same node associated with that condition. An example of a prefix tree is shown in Fig. 5.

Figure 5 Example of a prefix tree.

As both rules have the same first three conditions, they share the first three nodes.

Rule pattern searching

Before describing the rule pattern search, it is worth introducing a new definition and presenting a theorem that provides the theoretical motivation for the search algorithm.

Definition 2 Let E be an edge that connects two nodes x and y. Let S be a set of rules passing through E. We say that E is optimal if

(3) {(x,y)∈S×S:consequent(x)≠consequent(y)}=∅.

In other words, an edge is optimal only if it is connected to a single consequent. In Fig. 5, for instance, there are four optimal edges: A-B, B-C, A-C, C-B, while the remaining two edges are not optimal.

Theorem 1 Let P=(E1,…,EN) be a path, where {Ei}i=1 are edges. If the last edge EN is an optimal edge then P is an optimal path.

The proof of the theorem is given in Appendix A (Proof of Theorem 1). Now, it comes to the description of the search algorithm. The search cannot be exhaustive since the number of possible rule patterns in a fuzzy knowledge base can be very high, even huge. Therefore, it adopts a greedy strategy and it focuses only on extracting a subset of the rule patterns with the highest efficiency. The search algorithm uses the prefix tree as a decision tree, constructed in the previous stage of the RRC algorithm, and performs a left-to-right depth-first search (DFS) visit of the tree for each decision (Cormen, Leiserson & Rivest, 2009). The search algorithm is greedy because the attribute importance of nodes decreases from left to right. For each explored node in DFS, there are four possible cases: The current path, i.e., the rule pattern, is optimal. Therefore, it is added to the list of candidate rule patterns and the search is stopped since adding another edge, by Theorem 1, can only decrease the efficiency of the path.

The rule pattern is not optimal, and its efficiency is η>R (i.e., greater than the minimum efficiency), and the number of rules covered by the consequent is ν>C (i.e., greater than the minimum coverage). The path is added to the list of candidate rule patterns, and the search continues.

The rule pattern is not optimal, and its efficiency is η≤R, while ν≥C. The search continues but the path is not added to the list of candidate rule patterns.

The rule pattern is not optimal, its efficiency is η≤R, and it results ν≤C. In this case, the DFS visit stops and the rest of the tree is visited.

It should be emphasized that the search algorithm is not applied directly to the complete tree derived from the initial knowledge base, but rather, iteratively, to a sequence of its reductions, since the application of the search algorithm to the complete tree can become computationally intractable when the total number of attributes of the knowledge base is large. Therefore, for each iteration, a reduction of the complete tree is computed and the search algorithm is executed. The number of iterations is given by the input parameter T. The reduction of the full tree is simply obtained by randomly discarding some attributes from the decision table, and constructing a pruned prefix tree, as shown in Fig. 6.

Figure 6 Prefix tree pruning: (A) example of discarding attributes; (B) whole prefix tree; (C) pruned prefix tree after the discarding of two attributes.

Sorting rule patterns and building the minimum set of rules

The result of a rule pattern search is a list of candidate rule patterns, which is usually larger than the original knowledge base. Therefore, in this phase only a subset of candidate rule patterns is extracted that can cover all the decisions produced by the initial knowledge base and, at the same time, minimize the number of rules and conditions. This is done in two steps. In the first step, a greedy strategy is applied to sort the list of rule patterns in descending order according to their efficiency value. In the second step, the set S of the first rule patterns is extracted from the sorted list of rule patterns, covering all decisions produced by the initial knowledge base. The set S is the output of the RRC algorithm.

In some cases, the accuracy may not be 100%, namely the set S does not cover all the decisions produced by the initial knowledge base. However, suppose it is necessary to reach an accuracy of 100%. In that case, the rules, whose decisions are not covered by S, are included in S, so that this set covers the entire knowledge base.

A JavaScript implementation (Camastra et al., 2024) of the RRC algorithm is available at https://github.com/angelociaramella/IFKBS.

Computational complexity of rrc algorithm

This section discusses the computational burden of RRC, focusing only on the worst case. Since the RRC algorithm consists of five different stages, it is necessary to evaluate the complexity of each single stage to get the overall computational complexity.

The first stage builds the decision table. The computational complexity of this operation is O(MN), where M and N are the maximum number of attributes of a rule and the number of rules in the knowledge base, respectively. In the second stage, i.e., sorting attributes by their importance, the computation of the importance of the attributes (i.e., fuzzy relations) and their sorting are performed. The complexity of the first operation is still O(MN). Since the sorting of the attributes requires O(Mlog⁡M), the complexity of the stage is O(MN+Mlog⁡M).

The third stage requires the construction of the prefix tree, whose computational complexity is again O(MN).

The fourth stage, i.e., rule pattern search, requires that the rule pattern search be repeated T times. Each iteration involves a DFS visit of a reduced prefix tree. The iteration requires a complexity of O(MN), for the construction of the reduced prefix tree, and a complexity of O(MND), where D is the total number of decisions of the knowledge base. Therefore, the complexity of each iteration is O(MN+MND). Since each iteration is repeated T times, the complexity of the fourth stage of the RRC algorithm is O(T(MN+MND)), or more simply O(TMN(D+1)).

In the fifth stage, the candidate rule patterns are sorted and the first rule patterns covering the initial knowledge base are extracted.

The sorting requires O(TMN(D+1)log⁡TMN(D+1)), while the complexity of the extraction is O(N). Then, the complexity of the stage is O(TMN(D+1)log⁡TMN(D+1))+ O(N). Since the latter term is negligible w.r.t the former, the complexity of the stage is O(TMN(D+1)log⁡TMN(D+1)).

Therefore, the overall complexity of RRC is the sum of the complexities of its component stages, namely:

O(MN)+O(MN)+O(Mlog⁡M)+O(MN)+O(TMN(D+1))+O(TMN(D+1)log⁡TMN(D+1)),

and, since the first five terms are negligible w.r.t the last one, it can be written as O(TMN(D+1)log⁡TMN(D+1). Finally, it is worth noting that RRC has lower complexity than Ripper Algorithm, in terms of the number of rules N. Ripper Algorithm has complexity O(Nlog2N), while RRC has complexity O(Nlog⁡N).

Experimental results

The proposed algorithm has been validated on four UCI benchmarks, namely, mushroom (Schlimmer, 1987), breast-cancer (Michalski et al., 1986), nursery (Olave, Rajkovic & Bohanec, 1989), primary-tumor (Cestnik, Kononenko & Bratko, 1986)1 , and was compared with Ripper (Cohen, 1995), Part (Frank & Witten, 1998), Lem2 (Grzymala-Busse, 1992) and Evolutionary based (Gen & Lin, 2023) algorithms (Camastra et al., 2024). See https://github.com/angelociaramella/IFKBS for raw data and code (JavaScript). The benchmark characteristics are summarized in Table 3. The performance of the algorithms was measured in terms of the number of rules extracted, coverage, and accuracy. Coverage measures the percentage of the original knowledge base rules, covered by the set of rules generated by the compression algorithm. Accuracy measures the percentage of the correct classification of the set of rules generated by the compression algorithm.

Table 3 Benchmark data set characteristics: number of rules and number of attributes.

Benchmark	Number of rules	Number of attributes	
Mushroom	8,214	178,728	
Breast-cancer	286	2,574	
Nursery	12,960	103,680	
Primary-tumor	339	5,763	
TERA	6,215	26,959	

The results on the mushroom, breast cancer, nursery, and primary-tumor benchmarks are reported in Tables 4–7, respectively. Experimental comparisons, on the benchmarks, with other recent methods (Yang et al., 2021; Chen et al., 2023) are not reported since their results are not reproducible due to the unavailability of their implementation. Moreover, these methods cannot theoretically compared with RRC algorithm, since their computational complexities are not declared.

Table 4 Comparison results on mushroom benchmark by considering the estimated number of rules, number of attributes, the obtained accuracy and the coverage percentage.

Benchmark	Number of rules	Number of attributes	Accuracy	Coverage	
Ripper	8	12	100%	100%	
Part	13	20	100%	100%	
Lem2	8	78	100%	100%	
Genetic algorithm	2,769	4,325	100%	98.7%	
RRC	10	29	100%	100%	

Table 5 Comparison results on breast cancer benchmark by considering the estimated number of rules, number of attributes, the obtained accuracy and the coverage percentage.

Benchmark	Number of rules	Number of attributes	Accuracy	Coverage	
Ripper	17	56	90.2%	100%	
Part	54	149	91.9%	100%	
Lem2	124	574	98.5%	100%	
Genetic algorithm	1,042	3,325	98.6%	98.7%	
RRC	53	238	100%	100%	

Table 6 Comparison results on primary-tumor benchmark by considering the estimated number of rules, number of attributes, the obtained accuracy and the coverage percentage.

Benchmark	Number of rules	Number of attributes	Accuracy	Coverage	
Ripper	10	31	44.00%	100%	
Part	51	160	68.43%	100%	
Lem2	175	2,625	90.00%	90.00%	
Genetic algorithm	2,610	10,000	90.00%	100%	
RRC	162	1,793	84.20%	100%	

Table 7 Comparison results on nursey benchmark by considering the estimated number of rules, number of attributes, the obtained accuracy and the coverage percentage.

Benchmark	Number of rules	Number of attributes	Accuracy	Coverage	
Ripper	115	460	97.73%	100%	
Part	220	667	99.80%	100%	
Lem2	603	3,668	100%	93%	
RRC	16	43	89.20%	100%	

Finally, the RRC algorithm was evaluated on the TERA knowledge base (Camastra et al., 2014, 2015, 2024; https://github.com/angelociaramella/IFKBS.git). TERA is a fuzzy decision system for environmental risk assessment related to the cultivation of genetically modified plants. The TERA knowledge base consists of 6,215 rules and 26,959 attributes. Once applied to the TERA knowledge base, RRC extracted 2,562 rules and 9,319 conditions, whose compression rates are 59.8% and 65.4%, respectively, guaranteeing a 100% of both coverage and accuracy.

Conclusions

This article presented the Reducing Rules and Conditions (RRC) algorithm, a fuzzy rule base minimization algorithm based on rough set theory and a greedy strategy. Fuzzy set theory was used to evaluate the importance of attributes, while the greedy strategy was used to construct a prefix tree, which is employed as a decision tree traversed by a left-to-right depth-first search (DFS) visit. Several knowledge bases from UCI benchmarks were used to evaluate the behaviour of the proposed approach and to compare its performance with some competing methods from the literature. In any case, the results are promising, as they lead to more compact knowledge bases compared to the competitors or, at least, to higher coverage and accuracy percentages. The latter results are a prerequisite for building explainable decision support systems. Nevertheless, future work is needed to establish RRC as one of the reference methods for developing interpretable inference systems. In particular, emphasis will be placed on the theoretical background of the method (e.g., the search of diverse rule pattern efficiency measures), and the investigation of the explainability of the obtained fuzzy rules will also require more in-depth attention by comparing recent results in this context (Gallo et al., 2020).

Appendix

Proof of theorem 1

The proof of Theorem 1 is as follows.

Given a set of rules R, and given an optimal rule pattern P, whose antecedent A∗ is composed of a union of conditions (Aj)j=1Z, we can say that P covers S rules in R. If we add a new condition X to A∗, S cannot increase. Therefore, the efficiency P for the consequent D before adding X is:

(4) η(P,D)=|sup(P)∩R(P,D)|λ(P)K,

since the first term of the denominator is null, being P optimal.

After the addition of X to A∗ the efficiency of the new rule pattern P′ is:

(5) η(P′,D)=|sup(P′)∩R(P′,D)|((λ(P)+1)K,

Since both inequalities:

|sup(P′)∩R(P′,D)|≤|sup(P′)∩R(P′,D)| and ((λ(P)+1)K≥λ(P)K hold,

we conclude that:

(6) η(P′,D)≤η(P,D).

Since the addition of any condition to the optimal rule pattern decreases the efficiency of the rule pattern, the rule pattern P is optimal.

Part of the work was developed by Salvatore Sposato, under the supervision of Francesco Camastra and Antonino Staiano, as a thesis for M. Sc. in Applied Computer Science at the University of Naples Parthenope.

Additional Information and Declarations

Competing Interests

Author Contributions

Data Availability

1 Information about other benchmarks can be found at http://archive.ics.uci.edu/ml/datasets.php.

Angelo Ciaramella is an Academic Editor for PeerJ Computer Science.

Francesco Camastra conceived and designed the experiments, analyzed the data, prepared figures and/or tables, authored or reviewed drafts of the article, and approved the final draft.

Angelo Ciaramella conceived and designed the experiments, analyzed the data, prepared figures and/or tables, authored or reviewed drafts of the article, and approved the final draft.

Giuseppe Salvi conceived and designed the experiments, performed the experiments, analyzed the data, performed the computation work, prepared figures and/or tables, authored or reviewed drafts of the article, and approved the final draft.

Salvatore Sposato performed the experiments, analyzed the data, performed the computation work, prepared figures and/or tables, authored or reviewed drafts of the article, and approved the final draft.

Antonino Staiano conceived and designed the experiments, analyzed the data, prepared figures and/or tables, authored or reviewed drafts of the article, and approved the final draft.

The following information was supplied regarding data availability:

The data is available at GitHub and Zenodo:

- https://github.com/angelociaramella/IFKBS.git.

- Ciaramella, A. (2024). On the Interpretability of Fuzzy Knowledge Base Systems. In PeerJ in Computer Science. Zenodo. https://doi.org/10.5281/zenodo.13962705.

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
