# Peer review of "On the interpretability of fuzzy knowledge base systems"

_PeerJ Computer Science, doi:10.7717/peerj-cs.2558_

## Round 0.1 · original submission · Major Revisions

Thank you for submitting your manuscript to PeerJ Computer Science. After careful consideration of the reviewers' comments, we regret to inform you that major revisions are required before your paper can be considered further for publication.

The reviewers have raised several concerns regarding the methodology and experimental evaluation presented in your paper. These issues need to be thoroughly addressed to ensure the robustness and validity of your findings.

In addition, we recommend a comprehensive rereading of your manuscript to correct any typographical, grammatical, or formatting errors that may have been overlooked. Ensuring clarity and coherence in your writing will greatly enhance the readability and overall quality of your paper.

Please revise your manuscript accordingly and provide a detailed response to each of the reviewers' comments. We look forward to receiving your revised submission and appreciate your understanding and cooperation in improving the quality of your work.

**Language Note:** The review process has identified that the English language must be improved. PeerJ can provide language editing services - please contact us at [email protected] for pricing (be sure to provide your manuscript number and title). Alternatively, you should make your own arrangements to improve the language quality and provide details in your response letter. – PeerJ Staff

·

Basic reporting

In this paper, the Reducing Rules and Conditions (RRC) algorithm, an approach to fuzzy rule base
minimization based on rough sets theory and a greedy algorithm, is presented.
Fuzzy set theory was used to evaluate the importance of attributes, and the greedy algorithm was used to build a prefix tree that is used as a decision tree traversed by a left-to-right depth-first search (DFS) visit.

Authors claimed that the results are promising, as they lead to more compact knowledge bases compared to the competitors or, if not, to higher percentage coverage and accuracy.

But, the theoretical background of the method (e.g., efficiency measures and demonstration of optimal paths), and investigation of the explainability of the fuzzy rules obtained need to be more in-depth attention.

Also, the obtained results need to be compared to previous publishedworks and so can be verified.

Experimental design

The proposed algorithm has been validated on four "Outdated" UCI benchmarks, i.e., mushroom (Schlimmer, 1987), breast-cancer (Michalski et al., 1986), nursery (Olave et al., 1989), primary-tumor (Cestnik et al., 1986), and was compared with Ripper (Cohen, 1995), Part (Frank and Witten, 1998), Lem2 (Grzymala-Busse, 1992) algorithms.

Benchmark characteristics are summarized in Table 3. The performance of the algorithms have been measured in terms of number of rules extracted, Coverage, and Accuracy.

I recommend to use recent published works to comapre with.
Also, the methdology of the results need to be clairiefed by compare and contrast flowcharts with pseudocode and code, and how to use them effectively to discuss the strengths of the proposed algorithm and its development

Validity of the findings

The obtained results need to be compared to previous publishedworks and so can be verified.
The theoretical background of the method (e.g., efficiency measures and demonstration of optimal paths), and investigation of the explainability of the fuzzy rules obtained need to be more in-depth attention.
The methdology of the results need to be clairiefed by compare and contrast flowcharts with pseudocode and code, and how to use them effectively to discuss the strengths of the proposed algorithm and its development

Additional comments

The qulaity of figures need to be improved

Cite this review as

Reviewer 2 ·

Basic reporting

See Section 4.

Experimental design

Page 10: None of the results that are given for RRC can be replicated. The authors need to redo this section so that the results can be replicated.

Validity of the findings

See Section 4

Additional comments

This paper presents a method for the important problem of simplifying rules by establishing which attributes are important.

Page 2, line 48: “five steps” are mentioned, but Fig. 1 only shows three steps. Please reconcile this discrepancy.

Page 4, line 101: A section number is missing.

Page 4, Eq. (1): Please explain where (1) comes from or derive it.

Page 5, line 127: Please reword “we’re going to deepen in …”.

Page 6, line 153: A section number is missing.

Page 6: Building a prefix tree: It seems that the author’s prefix tree depends on the ordering of the attributes in the rules, but there is no natural ordering of attributes in rules. If the authors can demonstrate that their method gives the same results regardless of the ordering of the attributes, then this paper could be considered for revision. If they cannot do this, then this would be a fatal flaw for their method, and the paper should be rejected.

Page 7, Fig. 3: Please explain how the numerical significance numbers shown on these figures were obtained.

Page 7, Fig. 4: It would be better to use Fig. 3 items to demonstrate the prefix tree.

Page 8, Def. 2: To me, this definition is vague. Please provide an example of an edge that is not optimal.

Page 8, line 171: “optimal path” needs to be defined.

Page10, Experimental results: Please provide details for at least one RRC case, so that readers will really understand how to use your method.

Page 10: None of the results that are given for RRC can be replicated. The authors need to redo this section so that the results can be replicated.

Cite this review as

Reviewer 3 ·

Basic reporting

In the article, the authors introduce an approach to fuzzy rule base minimization based on rough sets theory.
The algorithm proposed by the authors is composed of five steps, and it requires three parameters: the minimum efficiency R, the minimum rule coverage C, the number of iterations T.
The aim of the algorithm is to identify and delete the attributes that are irrelevant in a set of fuzzy rules.
In the article, the authors test their algorithm on 5 real data sets. The algorithm has good results.

Although the algorithm steps are not always formally presented in mathematical terms, their descriptions and the illustrative examples given by the authors are sufficient to understand the approach.
Algorithm’s main step consists of traversing a graph representing the set of rules, in order to select relevant rule patterns according to the parameters R, C, T. In the end of the description of this step, the authors state that :
“It is worth stressing that the search algorithm is not directly applied on the complete tree derived from the initial knowledge base, but, iteratively, on a sequence of its reduction” […] The reduction of the complete tree is simply obtained by randomly discarding some attributes from decision table”.
It would be interesting if the authors could explain how many attributes are randomly eliminated (quantity), given that some of the datasets tested by the authors have a substantial number of attributes: Mushroom has 178728 attributes, breast-cancer: 2574, Nursery: 103680, and Primary-tumor: 5763.

In the experiments, it seems to me that the datasets are not split into two sets (one for trains and one for tests), (or else the authors have not mentioned this) while the authors measure the rate of correct classification? The algorithm must be evaluated on a test dataset.

Presentation: I recommend that the authors edit the article to remove the numerous typos it contains (introduction, section of the algorithm).

Experimental design

The authors test their algorithm on 5 real data sets (Mushroom, breast-cancer, nursery, primary-tutor, and TERA) and compare their results to well-known approaches (Ripper, Part, Lem2, Genetic algorithm). It seems to me that the datasets are not split into two sets (one for trains and one for tests).

Validity of the findings

The algorithm of the authors has good results.

Cite this review as

---

## Round 0.2 · Major Revisions

I hope this email finds you well. After a thorough review of your manuscript by the assigned reviewers, I would like to inform you that, while there is potential in your work, several significant concerns have been raised regarding the experimentation and methodology.

The reviewers have pointed out that certain aspects of the experimental setup lack sufficient clarity and justification. In particular, they believe that more detailed explanations and stronger validations are necessary to support your findings. Additionally, methodological improvements have been recommended to ensure the robustness and reliability of the results.

In light of these concerns, we are requesting major revisions to the manuscript. We kindly ask that you carefully address each of the reviewers' comments in your revised submission, providing additional detail and supporting evidence where necessary.

·

Basic reporting

In this paper, the Reducing Rules and Conditions (RRC) algorithm, an approach to fuzzy rule base
minimization based on rough sets theory and a greedy algorithm, is presented.
Fuzzy set theory was used to evaluate the importance of attributes, and the greedy algorithm was used to build a prefix tree that is used as a decision tree traversed by a left-to-right depth-first search (DFS) visit.

Authors claimed that they have explained these results with higher percentage coverage and accuracy.


Also, the obtained results had mentioned two articles in the previous published works and so they can be verified.

Experimental design

Authors have to be clairiefed by compare and contrast flowcharts with pseudocode and code, and how to use them effectively to discuss the strengths of the proposed algorithm and its development.

Validity of the findings

The methdology of the results need to be clairiefed by compare and contrast flowcharts with pseudocode and code, and how to use them effectively to discuss the strengths of the proposed algorithm and its development

Additional comments

Authors said that they had improved the qulaity of figures.

Cite this review as

Reviewer 2 ·

Basic reporting

Page 2, line 4: it would be very helpful to the readers for you to mention that the blocks of Fig. 1 are explained in “Reduce Rules and Condition Algorithm.”

Page 2, Eq. 1: In your reply to my earlier comment about where this equation comes from, you stated that “it cannot be derived by other equations but only justified. …” It is not at all obvious what each of the terms in this equation stands for. Please include an example to illustrate this equation and to demonstrate that it does what you claim.

Additionally, in mathematics, “sup” is short for “supremum.” Your use of it here for the set for rules that matches the antecedent of the rule pattern <script P> is confusing. I strongly suggest you use something other than “sup,” e.g. “sor”.

Page 2, line 129: Please explain how minimum efficiency and rule coverage can be an input before a rule base is even specified.

Note, also that symbol “C” has already been used in (2); so, another symbol should either be used in (2) or here.

Page 7, Fig. 3: Although you indicated in your reply to my request (that you explain how the numerical significance numbers shown on these figures were obtained) that they are computed by means of Eq. (1)”, I still do not understand how the two components of (1) are found from your earlier equations on Page 3, in which all sorts of things are defined and must somehow be computed. I must insist that you provide a numerical example in order to help readers use the results that are in your paper.

Experimental design

Page7, line 211, and the section “Experimental Results”: In my first review I stated: “Please provide details for at least one RRC case, so that readers will really understand how to use your method,” and “None of the results that are given for RRC can be replicated. The authors need to redo this section so that the results can be replicated.” You replied to these by referring to your Python code and its URL and stating that: “The readers can use immediately without it needing any further explanations or help.”

I downloaded this code and found: (1) The Read me file only had the name of the code; (2) The code has no explanations; (3) The code is in PYTHON, but all readers will not use PYTHON: (4) For “Scripts,” I got the following message: “Sorry about that, but we can’t show files that are this big right now. Consequently, the authors reply that “The readers can use immediately without it needing any further explanations or help” is not consistent with their code and other files.

Validity of the findings

Unless a reader is able to replicate the results, the validity of the findings cannot be confirmed.
See Section 2.

Cite this review as

Reviewer 3 ·

Basic reporting

The authors have sufficiently addressed the reviewers' comments and made the necessary revisions. The manuscript has improved in clarity and structure. Given these updates, I now recommend the paper for acceptance.

Experimental design

The experimental design is satisfying in the revised version of the article.

Validity of the findings

The authors' algorithm performs well on real datasets.

Cite this review as

---

## Round 0.3 · accepted · Accept

I hope this message finds you well. After carefully reviewing the revisions you have made in response to the reviewers' comments, I am pleased to inform you that your manuscript has been accepted for publication in PeerJ Computer Science.

Your efforts to address the reviewers’ suggestions have significantly improved the quality and clarity of the manuscript. However, I would like to invite you to address the last comments raised by reviewers reported below:

Reviewer Comment

The authors have addressed all of my comments from the last review. There are still a few things that need to be addressed:
1. Page 4/12, line 128: The last sentence, beginning with “In Figure 3” needs to be completed.
2. In Fig. 3, in the Efficiency equation, a term seems to be missing from the denominator, since Eq. (2) has three terms in its denominator.
3. Page 9/12, Experimental Results: The authors present results in Tables, but provide no discussions about any of them. Some discussion needs to be provided.

Thank you for your commitment to enhancing the paper. I look forward to seeing the final published version.

Reviewer 2 ·

Basic reporting

The authors have addressed all of my comments from the last review. There are still a few things that need to be addressed

Experimental design

No further comments.

Validity of the findings

No further comments.

Additional comments

The authors have addressed all of my comments from the last review. There are still a few things that need to be addressed:
1. Page 4/12, line 128: The last sentence, beginning with “In Figure 3” needs to be completed.
2. In Fig. 3, in the Efficiency equation, a term seems to be missing from the denominator, since Eq. (2) has three terms in its denominator.
3. Page 9/12, Experimental Results: The authors present results in Tables, but provide no discussions about any of them. Some discussion needs to be provided.

Cite this review as